# Position: Graph Learning Will Lose Relevance Due To Poor Benchmarks

Maya Bechler-Speicher [* 1]   Ben Finkelshtein [* 2]   Fabrizio Frasca [* 3]   Luis Müller [* 4]   Jan Tönshoff [* 4]
Antoine Siraudin [4]   Viktor Zaverkin [5]   Michael M. Bronstein [2 6]   Mathias Niepert [7 5]   Bryan Perozzi [8]
Mikhail Galkin [8]   Christopher Morris [4]

## Abstract

While machine learning on graphs has demonstrated promise in drug design and molecular property prediction, significant benchmarking challenges hinder its further progress and relevance. Current benchmarking practices often lack focus on transformative, real-world applications, favoring narrow domains like two-dimensional molecular graphs over broader, impactful areas such as combinatorial optimization, relational databases, or chip design. Additionally, many benchmark datasets poorly represent the underlying data, leading to inadequate abstractions and misaligned use cases. Fragmented evaluations and an excessive focus on accuracy further exacerbate these issues, incentivizing overfitting rather than fostering generalizable insights. These limitations have prevented the development of truly useful graph foundation models. *This position paper calls for a paradigm shift toward more meaningful benchmarks, rigorous evaluation protocols, and stronger collaboration with domain experts to drive impactful and reliable advances in graph learning research, unlocking the potential of graph learning.*

## 1. Introduction

Graphs are versatile mathematical structures capable of modeling complex interactions among entities across a wide range of disciplines, including the life sciences (Wong et al., 2023), social sciences (Easley & Kleinberg, 2010), and optimization (Cappart et al., 2021), underlining the need for specialized machine-learning methods to extract meaningful insights from graph-structured data. Hence, in recent years, *message-passing graph neural networks* (MPNNs) (Gilmer et al., 2017) have emerged as the leading architecture for machine learning on graphs. These architectures—and, more broadly, *graph neural networks* (GNNs)—have become prominent topics at top-tier machine learning conferences,[1] demonstrating promising performance across a diverse range of applications. Notable examples include their role in breakthroughs such as discovering new antibiotics (Stokes et al., 2020; Wong et al., 2023) and advancements in weather forecasting (Lam et al., 2023).

Despite these successes, we contend that for graph learning to remain *relevant* and *impactful*, current benchmarks need to be aligned with such truly *transformative real-world applications*. While various benchmarks have been proposed, many existing datasets focus on narrow domains or address problems with questionable practical relevance. For instance, popular benchmarks frequently feature two-dimensional molecular graphs (Hu et al., 2020a; Morris et al., 2020), neglecting critical three-dimensional geometric structures. Additionally, many studies report state-of-the-art results on (synthetic) datasets like ZINC (Dwivedi et al., 2022b), which lack sufficient (real-world) justification for their graph-based approach, further complicating their utility. Empirical studies in graph learning often suffer from methodological shortcomings. Inconsistent dataset splits and evaluation protocols across studies undermine the validity of comparisons, while the reliance on small datasets frequently results in high-variance outcomes with limited statistical significance. Due to these limitations and the scarcity of sufficiently large and diverse datasets, MPNNs and GNNs have shown limited evidence of scalability to large pre-trained or foundation models.

**Present work** In this position paper, we argue that graph learning must significantly revise its current datasets and benchmarking practices to remain impactful and relevant; see Figure 1 for an overview. Specifically, we

1. discuss the current shortcomings in graph learning

---
[*]Equal contribution  [1]Meta [2]University of Oxford [3]Technion - Israel Institute of Technology [4]RWTH Aachen University [5]NEC Laboratories Europe [6]AITHYRA [7]University of Stuttgart [8]Google Research. Correspondence to: Maya Bachler-Speicher <mayab4@mail.tau.ac.il>, Luis Müller <luis.mueller@cs.rwth-aachen.de>.

*Proceedings of the 42^nd International Conference on Machine Learning*, Vancouver, Canada. PMLR 267, 2025. Copyright 2025 by the author(s).

---
[1]http://tinyurl.com/mpn89vju

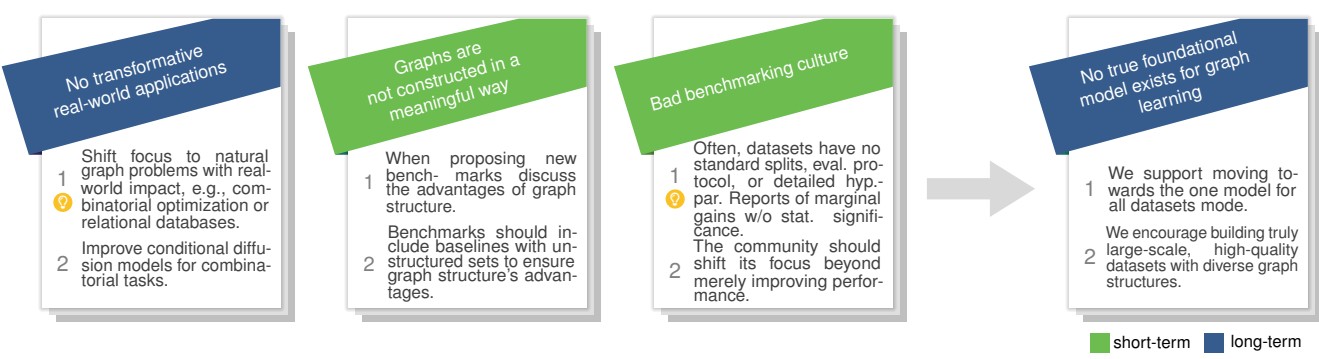

Figure 1. Overview of the current challenges in benchmarking for graph learning and possible remedies.

benchmarks, including the lack of transformative real-world problems, an overfocus on specific data modalities, and fragmented evaluation protocols, resulting in the absence of true foundation models for graph data;

2. propose possible remedies to address these shortcomings, offering actionable recommendations for the graph learning community; and

3. based on our assessment of current graph benchmarks, we tune various new baselines and reference models on molecular prediction tasks, large-scale heterophilic datasets, and study in- and cross-domain transfer in a pre-training/fine-tuning setup.

***Overall, in this position paper, we argue that the benchmarking aspect of graph learning requires a significant revision for the field to stay impactful and relevant, including the design of current datasets, the investigated data modalities, and current benchmarking practices.***

In the remaining part of this section, we provide a critical overview of the field's current state. In the following four sections, we highlight four current shortcomings of graph datasets and benchmarking practices and their possible remedies.

**Basic terminology**  Graph learning comprises several regimes. The most common ones are *graph-level* and *node-level predictions* (i.e., classification or regression). In the former, we are given a training set of graphs and aim to train a GNN to make meaningful graph-level predictions outside this training set. In the latter, we instead seek to make predictions for nodes in a given graph or set of graphs; the setup here is either *transductive* or *inductive*. In the transductive setting, we are given a single graph with a subset of the nodes being the training set, and we aim to train a model to make correct predictions for the nodes outside this training set. In the inductive setting, we are given a training set of graphs with node (class-)labels and aim to train a model to make correct predictions for the nodes of

unseen graphs. Similarly, we can define *edge-level* or *link prediction*. In addition, *graph generation* aims to generate graphs modeled to a given data distribution proxied via a training dataset.

**Related work**  One of the first efforts towards more principled benchmarking of GNNs was taken by Dwivedi et al. (2020), who proposed a suite of real and synthetic graphs spanning a variety of node-, edge-, and graph-level tasks as well as an attempt to standardize evaluation protocols. However, the majority of the tasks either have a graph structure *superimposed* on the original dataset (such as graphs extracted from vision datasets like CIFAR10, which have been solved in the vision community) or focus on small synthetic graphs with a saturated performance. Another limiting factor is the strongly suggested model size below 500k parameters that was supposed to test models' *inductive biases*. While reasonable for the state of graph learning in 2020, such a manually set parameter count ceiling makes little sense in modern deep learning, where scaling laws suggest model capabilities grow with both dataset size and parameter count (Hoffmann et al., 2022; Schaeffer et al., 2023; Wei et al., 2022).

Soon after, Hu et al. (2020a) released the *Open Graph Benchmark* (OGB), a comprehensive suite of datasets encompassing various domains, tasks, and graph distributions. The authors proposed to gather results in a centralized, publicly visible leaderboard. The submission system requires researchers to provide test results, the corresponding validation performance, the number of learnable parameters, and some information about the tuning procedure. This effort goes in the direction of more informative and standardized benchmarking practices. Nevertheless, many datasets in the suite (such as 2D molecular graphs or academic citation networks) are still far from transformative real-world applications. As we discuss later in Sections 2 and 3, these graphs either fail to encode relevant information (e.g., 3D spatial arrangements of atoms) or induce a structural inductive bias

that is of unclear advantage for downstream generalization performance. While we note that some (large-scale) more impactful benchmarks are exposed by OGB, the research community has focused on them with relatively lower priority. This is likely due to the inherent difficulty of scaling more sophisticated and expressive architectures to larger graphs or the interest drawn by more specific settings, such as heterophilic networks, not generally covered by OGB.

Dwivedi et al. (2022b) proposed benchmark datasets to assess the long-range capabilities of GNNs, also transforming computer vision datasets into graph datasets, empirically showcasing the benefits of *graph transformers* (GTs) (Müller et al., 2024) over MPNNs. However, Tönshoff et al. (2024) have shown that the reported performance gap of graph transformers on these tasks is overestimated due to suboptimal hyperparameter choices, showcasing improper benchmarking practices. In addition, Errica et al. (2020) proposed a more meaningful evaluation protocol for GNNs; however, their efforts primarily focused on the small datasets from Morris et al. (2020).

Recently, Coupette et al. (2025) introduced a formal framework to assess the quality of graph learning datasets, devising two complementary measures. In addition, they conducted extensive experiments and proposed recommendations for improving benchmarking practices in graph learning.

In 2D graph generation, many papers still evaluate on QM9 (Wu et al., 2018) or ZINC250K (Gómez-Bombarelli et al., 2018) even though these datasets are regarded as solved, i.e., most state-of-the-art models obtain near-perfect performance. In addition, the widely used SPECTRE benchmark (Martinkus et al., 2022) is also saturated, and results are not consistently reported across papers.

Nickel (2024) showed analytically that for widely considered inference settings in complex social systems, including graph learning, the train-test paradigm does not only lack justification but is indeed invalid for any risk estimator, including counterfactual and causal estimators. These formal impossibility results highlight a fundamental epistemic issue in graph learning, i.e., that for many tasks we cannot know how good our models really are under current data collection practices.

See Appendix A for an extended discussion of related work.

## 2. Missing transformative real-world applications and supporting benchmarks

We believe that the graph learning community has not yet identified benchmarks showcasing transformative real-world applications that genuinely exploit the benefits of machine learning on graphs. Unlike the computer vision

or natural language domains, graph learning has no "natural" application areas, as graphs usually abstract other data modalities featuring more or less evident relational structures.

In the past, graph learning primarily focused on benchmarking newly developed GNN architectures on datasets stemming from specific applications. The molecular domain, e.g., predicting properties of small 2D molecular graphs (Hu et al., 2020a; Morris et al., 2020) has been an area of particular interest. Meanwhile, meaningful small 2D molecular graphs only cover minor, niche sub-fields in chemistry or drug discovery, where it is more natural to relate a 3D structure and a property evaluated at a quantum mechanical level of theory. In addition, transforming raw chemical data obtained from, e.g., experiments or quantum mechanical calculations into 2D molecular graphs can be time-consuming; it often results in the loss of important information and, thus, fails to capture the relationship between spatial atomic arrangements and properties.

In addition to challenges in supervised graph learning, similar issues arise in graph generation. Most papers benchmark their methods using 2D molecular graph generation (Vignac et al., 2023). However, 3D point clouds might be better suited and preferred by domain experts for such tasks, as the geometric structure of molecules is crucial for real-world applications, such as molecular docking or fragment linking (Igashov et al., 2024; Schneuing et al., 2024). For example, despite its prominence, DiGress (Vignac et al., 2023)—one of the most cited works in graph generation over the past two to three years—has seen few practical follow-ups. Notably, most citations serve as background rather than extensions of their ideas. The utility of generating structured data remains unclear—with evaluating the quality of generated graphs without ground-truth data being one of the key challenges (Handa et al., 2023)—leaving this field without a clear application-driven focus. As a result, critical topics, such as generating graphs with strong structural constraints or scaling methods to large graphs, receive limited attention.

**Suggested remedies** The community should shift focus from smaller, less relevant 2D molecular benchmarks to problems naturally represented as graphs. One promising area is combinatorial optimization, where graphs encode problem instances, such as in the vehicle routing problem (Toth & Vigo, 2002) or bipartite graphs in integer-linear programming (Schrijver, 1986), as discussed in Cappart et al. (2021). Combinatorial optimization benchmarks offer distinct advantages: (1) clear real-world applications, (2) easy generation of large datasets, and (3) ideal testbeds for studying size generalization.

Beyond combinatorial optimization, other high-potential areas for GNNs include satisfiability solving (Biere et al.,

2021), recommender systems (Wu et al., 2022), social networks (Newman, 2003), and power-flow networks (Owerko et al., 2020). Projects like RelBench (Robinson et al., 2024) and 4DBInfer (Wang et al., 2024) demonstrate GNNs' utility in automating machine learning on relational databases, while TpuGraphs (Phothilimthana et al., 2023) highlights their potential in computer systems. GNNs are also effective in automated chip design, such as in AlphaChip, where reinforcement-learning-based models leverage netlist embeddings (a hypergraph of circuit components and their connections) (Mirhoseini et al., 2021; 2024).

Industrial datasets like social networks often involve sensitive data, limiting accessibility. Better anonymization methods could address this, such as generating anonymous graphs similar to real-world data (Yoon et al., 2023). While GNNs have been used for de-anonymization (Crețu et al., 2022), anonymized graph generation remains an open challenge.

Graph generation also holds promise for design-related tasks, mainly through diffusion models (Liu et al., 2024). For example, these models create heat maps for sampling solutions to combinatorial problems (Li et al., 2024; Sun & Yang, 2023). While competitive, further work is needed to improve efficiency and understand GNN capabilities in this context. We hypothesize that for any prediction task $p(y|x)$, the conditional generation counterpart $p(x|y)$ is also valuable, provided $y$ is easy to evaluate or $p(y|x)$ is robust enough to assess generated samples reliably. By prioritizing combinatorial optimization and graph-representable problems, the community can advance theoretical insights and practical applications, providing a more straightforward path to real-world impact.

## 3. Graphs are not necessarily constructed in a meaningful way

As discussed above, graphs are higher-level abstractions of real-world phenomena or observables featuring relational structure. Hence, their effectiveness in tackling a specific task will inherently depend on how they are constructed and whether the relational information they encode predicts the problem (Halcrow et al., 2020). However, commonly adopted graph learning benchmarks often do not consider the meaningfulness, relevance, and completeness of the proposed constructed graphs; in fact, they sometimes either represent unsuitable formalisms for the data modality at hand, fail to encode important information, or do not correlate with the considered learning targets. We provide some examples in the following.

A first exemplary case is that of the PASCALVOC-SP and COCO-SP datasets (Dwivedi et al., 2022b). Their graphs encode coarse-resolution images, with rag-boundary edges drawn to connect super-pixels corresponding to segmented regions. However, this modeling choice is not grounded in theoretical or empirical justification. As such, it is unclear whether modeling images as graphs in this way is helpful for object detection or the vision domain in general.

Spatiotemporal datasets, such as traffic networks, e.g., PEMS-BAY and METR-LA (Li et al., 2017), or air quality measurements. e.g., AQI (Zheng et al., 2015), rely on sensor readings taken at various locations. Subsequently, a thresholded Gaussian kernel is applied to the pairwise distances between these sensor locations to construct the graph structure, introducing structure to an otherwise fully connected weighted graph by imposing a threshold to decide which connections are retained. While this preprocessing step provides a relational structure that facilitates using graph-based methods, it is fundamentally arbitrary and may misrepresent the system's dynamics. For instance, the choice of the threshold value is often heuristic, potentially omitting meaningful connections, e.g., emphasizing short-range interactions, which may or may not be the right choice for the problem. This highlights the need for more principled, data-driven methodologies for constructing spatiotemporal graphs.

Again, another relevant example is represented by the widely adopted ZINC benchmark (Dwivedi et al., 2020). ZINC contains small molecular graphs, whereby nodes represent atoms and edges the chemical bonds between them. This form of relational structure captures natural chemical information. Still, nodes and edges are attributed solely to the type of atoms and chemical bonds they represent, missing encoding important structural information such as the 3D atom coordinates and the SMILES-derived features that are easily obtained via software packages such as RDKit (Landrum, 2016).

In some other compelling cases, relational information can be natural to consider but not necessarily informative to solve the prediction task at hand. In particular, this issue has been studied in recent work by Bechler-Speicher et al. (2024). The authors show settings where MPNNs overfit to spurious correlations in the structure, whereas set-based models (Zaheer et al., 2017) only process node features and exhibit better generalization performance. Exemplary settings of this kind are those of citation networks, where nodes represent scientific articles connected by edges whenever one cites the other. This form of relational structure is exceptionally reasonable but not necessarily predictive for any task instantiated on these graphs. That is, textual similarity in the content could, e.g., better correlate with article category than simply patterns of citations.

**Suggested remedies** Virtually any real-world phenomenon and system can potentially be modeled as a

"graph" (Veličković, 2023), but this does not imply that any choice of relational structure is equally relevant or predictive or that a relational framework is a convenient modeling choice. Benchmarks should be designed in a way that accounts for these aspects systematically and quantitatively, openly, and structurally, considering the motto:

*"Not everything that could be modeled as a graph should be modeled as a graph."*

When proposing a new benchmark, authors should discuss the advantages of adopting a "relational" modeling framework, articulating the benefits expected from processing data framed in graphs w.r.t. other possible modalities. In addition, they should discuss the choice of node and edge features and the rationale for how edges are determined in the first place. Crucially, authors should not only illustrate *how* graphs are constructed but expand on *why* the chosen approach is expected to be advantageous for the prediction task at hand.

Quantitatively, we advise that benchmarks should always be accompanied by (adequately tuned) baselines such that comparisons with them will allow us to underscore the advantages of the considered structural information on generalization performance. Concretely, benchmarks should always report the performance of baselines that only process unstructured sets of node features,[2] or graphs whose connectivity is obtained solely from the similarity thereof. Benchmark guidelines should explicitly promote the quantifying performance of proposed approaches in relative terms w.r.t. these.

## 4. Bad benchmarking culture

We believe inadequate benchmarking culture significantly hinders the graph learning community, irrespective of impactful applications (see Section 2) or the usefulness of underlying graphs (see Section 3). While poor benchmarking exists across machine learning (Herrmann et al., 2024), it is particularly problematic in graph learning. Even for standard datasets (Morris et al., 2020), inconsistent evaluation protocols and dataset splits result in highly variable performance reports (see Appendix B.4), with some papers overestimating performance by reporting validation metrics (Errica et al., 2020). Small datasets like MUTAG (Morris et al., 2020), with only 188 graphs, lead to large standard deviations and unreliable comparisons, while some suffer from misclassifications or insufficient class representation (Li et al., 2023; Platonov et al., 2023).

Newly proposed architectures are often unfairly compared

---

[2]These could be, for example, instantiated as DeepSets (Zaheer et al., 2017) or transformer-based architectures (Müller et al., 2024).

to outdated baselines, with hyperparameters fine-tuned on a small number of datasets but not for baselines. Theoretically motivated GNNs (Maron et al., 2019; Morris et al., 2019) frequently claim inflated performance gains by avoiding comparisons with state-of-the-art models.

The community often overlooks the relevance of minor improvements. For instance, ZINC (Dwivedi et al., 2020) tasks can be easily solved with standard chemoinformatics tools (Landrum, 2016), yet incremental improvements on such benchmarks are often highlighted. Additionally, limited molecular and material modeling domain knowledge prevents meaningful task understanding. For example, current state-of-the-art models often ignore critical relationships between 3D structure and molecular properties.

In 2D graph generation, datasets like QM9 (Wu et al., 2018) and ZINC250K (Gómez-Bombarelli et al., 2018) dominate despite near-perfect performance. More robust benchmarks, such as MOSES (Polykovskiy et al., 2020) and GUACAMOL (Brown et al., 2019), remain underutilized due to high computational demands. Benchmarking inconsistencies, such as differing dataset splits (Siraudin et al., 2024), inappropriate reliance on novelty for QM9 (Vignac & Frossard, 2021), and inconsistent FCD reporting further exacerbate the issue. Current benchmarks emphasize unconditional generation, whereas real-world applications require conditional generation, which remains underexplored due to oversimplified tasks and strong baselines (Tripp et al., 2021).

Beyond molecules, benchmarking for graph generative models is even less standardized. Some studies rely on limited datasets like CORA or the SPECTRE benchmark (Martinkus et al., 2022), focusing on specific graph types but often omitting metrics like VUN and error bounds. Benchmarking for large graphs faces additional challenges due to the scarcity of practical datasets and even poorer standardization practices.

**Suggested remedies** The graph learning community must develop practical tasks and robust evaluation frameworks to address these challenges. Unlike LMsys Arena's ELO-based evaluation (Zheng et al., 2023), graph learning lacks trusted benchmarks resistant to manipulation. While domain expertise poses challenges, creating expert-validated benchmarks can significantly improve model evaluation and adoption.

A Kaggle-like competition with hidden test sets at the NeurIPS benchmark track could realistically assess models across domains like molecular prediction and combinatorial optimization. Addressing data quality issues requires larger, domain-relevant datasets such as ADMET BENCHMARK GROUP (Swanson et al., 2023) or PUBCHEMQC PM6 (Nakata et al., 2020), which provide diverse, real-

world data. Multidisciplinary collaboration is essential for curating datasets and translating real-world problems into graph learning tasks (You et al., 2020).

For 2D molecule generation, benchmarks like MOSES and GUACAMOL should replace outdated ones like QM9 and ZINC250K for serious evaluations. Future efforts must focus on computational efficiency and improved benchmarks such as SPECTRE, incorporating larger datasets with diverse structural properties. Evaluations must include error bars and report ratios and prioritize combined metrics like MMD and VUN. We advocate for new benchmarks extending existing frameworks to evaluate diverse, complex structures effectively.

## 5. Implication: No true foundation model exists for graph learning

In deep learning, large pre-trained foundation models (Llama Team, 2024; Gemini Team, 2024) that unify multiple modalities (e.g., text, images, video, audio) excel at predictive and generative tasks, reshaping research and industry. However, similarly impactful *graph foundation models* (GFMs) are yet to emerge. Domain-specific graph-based models have appeared recently (Mao et al., 2024) for tasks such as node classification (Zhao et al., 2024), neural algorithmic reasoning (Ibarz et al., 2022), knowledge graph reasoning (Galkin et al., 2024a;b), and molecular property prediction (Kläser et al., 2024; Sypetkowski et al., 2024). Yet, their performance often shows only marginal gains over standard supervised GNNs (Zhao et al., 2024; Kläser et al., 2024; Chen et al., 2024).

As argued in Sections 2 to 4, training on small datasets or academic tasks without rigorous evaluations hampers progress in graph learning. Additional challenges include: (1) differing symmetries and expressivity requirements across tasks (e.g., labeling trick GNNs (Zhang et al., 2021; Zhu et al., 2021) excel in link prediction but not node- or graph-level predictions); (2) learning representations for graphs with varying scales and feature spaces necessitates new strategies for *graph tokenization* and defining a universal *graph vocabulary* (Mao et al., 2024); (3) graph data availability is orders of magnitude smaller than text data, and a *token* for graphs lacks clear definition; (4) limited commercially relevant GFM applications, as discussed in Section 2.

**Suggested remedies** Despite these challenges, GFMs and robust real-world graph benchmarking are critical for advancing graph learning alongside progress in other deep learning areas. We propose shifting from *one model for one dataset* to *one model for all datasets* to provide a comprehensive view of model performance across diverse graphs. For example, instead of training a separate model for each task in a five-task benchmark, training one model for all

tasks is preferable. For particularly non-trivial setups (e.g., combining classification with regression), we suggest an *encoder-processor-decoder* approach (Battaglia et al., 2018; Ibarz et al., 2022): pre-train a unified backbone model and fine-tune task-specific encoders and decoders. Finally, we advocate for creating large-scale, high-quality datasets of diverse graph structures (e.g., sparse, dense, homophilic, heterophilic, directed, multi-relational), addressing data gaps with synthetic data (Palowitch et al., 2022), and ensuring data decontamination by excluding known test sets from pre-training corpora.

## 6. Alternative Views

Fields adjacent to graph learning, such as *geometric deep learning* (GDL) (Bronstein et al., 2017; 2021), are thriving and achieving remarkable successes. GDL has driven advancements in structural biology (Abramson et al., 2024; Jumper et al., 2021; Townshend et al., 2021) and materials science (Merchant et al., 2023; Reiser et al., 2022; Zeni et al., 2025). It also underpins state-of-the-art interatomic potentials for atomistic simulations at first-principles accuracy (Batatia et al., 2022; Batzner et al., 2022; Gasteiger et al., 2020; Hu et al., 2021b; Musaelian et al., 2023; Park et al., 2021; Schütt et al., 2017; Schütt et al., 2021; Simeon & Fabritiis, 2023; Thomas et al., 2018; Zaverkin et al., 2024), including the trend toward universal interatomic potentials (Batatia et al., 2023; Chen & Ong, 2022; Devereux et al., 2020; Smith et al., 2017; Kovács et al., 2023). Many GDL models leverage graph structures, though these are often constructed in task-specific or heuristic ways, such as using distance-based thresholds or graph sparsification techniques. Notably, graph sparsification is empirically crucial in both interatomic potential models and the Nobel Prize-winning ProteinMPNN, where sparse message passing plays a central role. While these constructions may not always be grounded in traditional graph semantics and often rely on heuristics, they have proven empirically effective. This challenges our earlier assertion that when graphs are not constructed in a meaningful way (Section 3), model performance and progress are likely to suffer.

## 7. Empirical evidence

Here, we support our claims made in the previous four sections with empirical evidence[3].

### 7.1. Graphs not necessarily constructed in a meaningful way

In Section 3, we raised concerns about the lack of correlation between graph structures in commonly used bench-

---

[3]Our code is available at `https://github.com/benfinkelshtein/PP-Benchmarks`.

*Table 1.* Comparison of different GNNs over OGB datasets, when using DeepSets (no graph), the original graph (Orig.) and a fixed expander graphs (Cayley).

| Model | MOLHIV | MOLBBBP | MOLBACE |
|---|---|---|---|
| Deepset (Empty) | $63.78_{\pm1.05}$ | $64.90_{\pm0.72}$ | $51.76_{\pm2.85}$ |
| GraphConv Orig. | $\mathbf{68.24_{\pm1.77}}$ | $\mathbf{64.11_{\pm4.50}}$ | $63.18_{\pm4.56}$ |
| GraphConv Cayley | $\mathbf{67.91_{\pm0.75}}$ | $61.60_{\pm4.48}$ | $56.94_{\pm7.50}$ |
| GIN Orig. | $\mathbf{69.65_{\pm2.58}}$ | $\mathbf{66.73_{\pm1.27}}$ | $53.44_{\pm4.52}$ |
| GIN Cayley | $\mathbf{68.61_{\pm1.40}}$ | $58.35_{\pm4.01}$ | $\mathbf{56.94_{\pm12.40}}$ |
| GAT Orig. | $\mathbf{67.21_{\pm1.30}}$ | $\mathbf{66.62_{\pm1.14}}$ | $53.21_{\pm1.34}$ |
| GAT Cayley | $\mathbf{67.80_{\pm3.45}}$ | $60.31_{\pm2.47}$ | $\mathbf{62.75_{\pm4.76}}$ |

*Table 2.* Evaluation results on the validation split of PCQM4Mv2.

| | Model | Val. MAE | #Param. |
|---|---|---|---|
| GNNs | GCN (Hu et al., 2021a) | 0.1379 | 2.0M |
| | GINE (Hu et al., 2021a) | 0.1195 | 3.8M |
| Transformers | TokenGT (Kim et al., 2022) | 0.0910 | 48.5M |
| | GRPE (Park et al., 2022) | 0.0890 | 46.2M |
| | Graphormer (Shi et al., 2022) | 0.0864 | 48.3M |
| | GPS (Rampášek et al., 2022) | 0.0858 | 19.4M |
| | ET (Müller et al., 2024) | 0.0832 | 16.8M |
| Ours | GINE | $0.0913_{\pm0.0002}$ | 22.7M |
| | GINE+RWSE | $0.0898_{\pm0.0001}$ | 22.7M |

marks and the intended learning targets. In this subsection, we provide empirical evidence to support this claim further. Recently, Deac et al. (2022); Wilson et al. (2024) proposed a message-passing scheme in which, during every odd layer, the original graph is disregarded in favor of propagating information through a fixed-structure expander graph—specifically, a Cayley graph. Ablation studies presented in Wilson et al. (2024) on multiple TUDATASET benchmarks showed that using the Cayley graph exclusively, without incorporating the original graph at any layer, sometimes improved performance. This finding is striking, as the Cayley graph does not inherently encode task-relevant information. These results align with the observations of Bechler-Speicher et al. (2024), who showed that making graphs more regular consistently improved performance. To further substantiate these findings, we replicate these experiments using the OGB graph-level benchmarks, strengthening the evidence for these observations. We also evaluate a DeepSets (Zaheer et al., 2018) baseline, where we drop the graph structure from the data, and therefore, the MPNN acts on an empty graph. Due to space limitations, the Appendix provides all the experimental details, including dataset information and tuned hyper-parameters.

The ROC-AUC scores average over 3 random seeds are summarized in Table 1. The best-performing model within the standard deviation range is marked in bold for each dataset. Across 5 out of 9 experiments, the non-informative regular Cayley graph outperformed or matched the performance of the original graph. Notably, for the MOLBBBP datasets, training with GraphConv achieved the highest AUC-ROC when the graph structure was dropped entirely.

### 7.2. Reassessing simple baselines on PCQM4Mv2

In Section 4, we discussed problematic practices of empirical evaluations of novel GNN architectures. One common issue is the citation of old, outdated reference results to quantify the performance improvements of new architectures over simpler baselines. Often, these baseline results

suffer from suboptimal hyper-parameters and are cited as-is for many years without reevaluation. As a consequence, the performance gains of newer architectures are commonly overestimated.

Here, we demonstrate this issue on the commonly used PCQM4Mv2 dataset (Hu et al., 2021a), among the few large-scale datasets for graph-level learning tasks, and is particularly popular for demonstrating performance improvements of graph transformers over simpler MPNNs. Experimental evaluations on this dataset typically cite the results for GCN (Kipf & Welling, 2017) and GINE (Xu et al., 2018a; Hu et al., 2020b) that were initially reported on the leaderboard of the 2021 OGB-LSC competition (Hu et al., 2021a) to represent standard MPNNs. These results suggest a validation MAE of around 0.12, while graph transformers commonly achieve MAEs below 0.09, indicating a substantial error reduction of over 25%.

We aim to reevaluate this performance difference by reassessing the reference results for standard message-passing GNNs. Specifically, we measure GINE's performance after re-tuning hyper-parameters for a larger 20-layer model with approximately 20 million parameters. We base our experiments on the same GINE architecture, with edge features used to obtain the original results. We make minor adjustments to align the setup with current deep learning practices used for transformers. Full details, hyper-parameters, and tuning budgets are reported in Appendix B. We report performance for GINE models with and without additional RWSE node features (Dwivedi et al., 2022a), also used by graph transformers such as GPS (Rampášek et al., 2022).

Table 2 provides the results of our evaluation. Even without additional RWSE features, the error of GINE drops by over 20% to 0.0913 simply by tuning the model configuration. When using additional structural features, the performance improves further to 0.0898, which is competitive with several graph transformer baselines. We do not claim that additional tuning could not further enhance the graph transformers' results. Instead, our results show how brittle empirical evaluations of GNNs generally are and that

the numbers reported throughout the literature often do not capture the actual progress of model capability or the lack thereof.

## 7.3. The meaningfulness of architectural changes

*Table 3.* Comparison of baseline GNNs with and without architectural modifications on heterophilous datasets.

| | Model | ROMAN-EMPIRE | AMAZON-RATINGS | MINESWEEPER |
|---|---|---|---|---|
| GCN | Re-evaluated | $44.41_{\pm 0.81}$ | $44.30_{\pm 0.52}$ | $72.90_{\pm 1.29}$ |
| | Reported | $73.69_{\pm 0.74}$ | $48.70_{\pm 0.63}$ | $89.75_{\pm 0.52}$ |
| SAGE | Re-evaluated | $80.80_{\pm 0.52}$ | $43.35_{\pm 0.80}$ | $83.76_{\pm 0.71}$ |
| | Reported | $85.74_{\pm 0.67}$ | $53.63_{\pm 0.39}$ | $93.51_{\pm 0.57}$ |
| GAT | Re-evaluated | $51.05_{\pm 0.90}$ | $44.52_{\pm 0.48}$ | $74.37_{\pm 0.94}$ |
| | Reported | $80.87_{\pm 0.30}$ | $49.09_{\pm 0.63}$ | $92.01_{\pm 0.68}$ |
| | Avg. % Gain | +43.56% | +14.63% | +19.49% |

In response to Section 4, we further exemplify the GNN evaluations' brittleness in the node-prediction setting. Platonov et al. (2023) proposed a set of heterophilous graph datasets to evaluate the performance of various GNNs, including both baseline and heterophily-specific GNNs. In their experiments, the authors used the official implementations of the heterophily-specific GNNs. However, they modified the baseline GNNs by adding a two-layer MLP after each neighborhood-aggregation layer. While mentioning that this architectural enhancement significantly improved baseline performance, the authors did not explore its impact further.

This enhancement raises several critical concerns regarding the validity of the comparisons: (1) No parameter budget was enforced, potentially leading to models of varying capacities. (2) The evaluation of the baseline GNNs followed a uniform protocol, whereas heterophily-specific GNNs were assessed using their respective codebases, which may incorporate diverse architectural choices and introduce unfairness. However, most important of all, (3) the significance of the specific baseline GNN protocol—including the two-layer MLP after each graph neighborhood aggregation, a linear encoder, a linear decoder, and GeLU activation—was acknowledged but not thoroughly analyzed, leaving its contribution to performance improvement unclear.

These issues render the performance comparisons in Platonov et al. (2023) less meaningful. Although the proposed datasets may serve as valuable benchmarks for heterophilous graphs, their utility cannot be conclusively determined without an evaluation protocol. This underscores the necessity of such a protocol (see Section 4)—even the most promising benchmarks require well-defined evaluation guidelines to assess their quality reliably.

To validate these concerns, we re-evaluated the baseline GNNs (GCN (Kipf & Welling, 2017), SAGE (Hamilton et al., 2017), and GAT (Velickovic et al., 2017)) using a fresh codebase that adhered to the hyper-parameters reported in

(Platonov et al., 2023) but excluded specific architectural modifications. Specifically, our evaluation omitted the linear encoder, the two-layer MLP after each aggregation layer, and the linear decoder and replaced GeLU activations with the standard ReLU.

As shown in Table 3, these architectural changes introduced in Platonov et al. (2023) resulted in significant average baseline performance gains of +43.56%, +14.63%, +19.49% on the roman-empire, amazon-ratings, and minesweeper datasets, respectively. This analysis does not question the validity of the proposed benchmarks but highlights the critical need for accompanying evaluation protocols. Such protocols should include a fixed model size limit to ensure fair parameter budgets and clear guidelines on allowable architectural modifications across all GNN layers.

## 7.4. Multi-task pre-training with encoder-processor-decoder

*Table 4.* Test performance on the upstream datasets, both trained in a single-task (ST) and multi-task (MT) setting, as well as random baselines (RD), on a single random seed.

| | Model | COCO-SP | MALNETTINY | PCQM4Mv2 |
|---|---|---|---|---|
| | | F1 ↑ | Acc. ↑ | MAE ↓ |
| RD | MPNN | 0.0002 | 23.10 | 5.2340 |
| | GT | 0.0005 | 19.60 | 5.2483 |
| ST | MPNN | 0.0817 | 81.80 | 0.1104 |
| | GT | 0.2947 | 81.90 | 0.1009 |
| MT | Empty | 0.0119 | 20.00 | 0.3915 |
| | MPNN | 0.0413 | 83.20 | 0.1363 |
| | GT | 0.1137 | 88.90 | 0.1441 |

In this section, as suggested in Section 5, we run a series of experiments to investigate multi-task pre-training/fine-tuning using an encoder-processor-decoder framework. Similar settings have been explored recently for small molecules in Kläser et al. (2024); Sypetkowski et al. (2024); Frasca et al. (2024). Here, we also want to study a cross-domain setting with data from vision, function-call graphs, large molecules, and social networks. The aim is to gather an initial signal on the suitability of this architectural pattern when pre-trained on a mix of vastly different graph tasks, even on a relatively small scale. As highlighted in Section 5, a large-scale, curated pre-training corpus is currently lacking, and we believe that positive results from our experiments could catalyze the community's efforts in building such a corpus, accompanied by standardized pre-training setups and evaluation procedures.

**Architectures, training, and evaluation** We train domain-specific encoders (e.g., embedding atom and bond types in molecules) and task-specific decoder MLPs. For the processor network, we both evaluate an MPNN based

on the GINE architecture (Xu et al., 2018b) and a GT based on Graphormer (Ying et al., 2021) with a soft attention bias and RWSE structural encodings (Dwivedi et al., 2022a). For our experiments, we assemble two sets of datasets: the *upstream mix* used for pre-training and the *downstream mix* used for fine-tuning. We freeze the processor weights during fine-tuning and learn a new encoder and decoder. If the downstream task permits, we reuse the encoder from one of the datasets in the upstream mix. In addition, for each upstream or downstream dataset, we train baseline models with the same model architectures as our pre-trained models but trained from scratch in a single-task fashion. All experimental details are enclosed in Appendix B.3.

**Upstream mix**  Our upstream mix contains PCQM4Mv2 (Hu et al., 2021a), COCO-SP (Dwivedi et al., 2022b), and MALNETTINY (Freitas et al., 2021). These datasets are diverse in various aspects, such as the underlying application domain, the type of prediction task, and graph size and sparsity. As a result, we do not expect a strong transfer between any of these tasks during pre-training. Rather, we select this upstream mix to investigate whether our MPNN and GT models can learn *general graph representations* useful for multiple, potentially unrelated tasks. To further support this assessment and quantify the benefits of learning a general-purpose processor network, we train an additional "empty graph baseline" in the multi-task setup, where the processor network is set to identity, and, hence, the graph structure is ignored. See Table 4 for our results. We observe that the MPNN and the GT show non-trivial performance compared to the empty graph baseline and their randomly initialized (untrained) counterparts on all three tasks. On COCO-SP and PCQM4Mv2, they both fall short of their single-task baseline. Surprisingly, on MALNETTINY, we find that MPNN and GT improve over their respective single-task performance when trained in a multi-task setting.

**Downstream mix**  Next, we evaluate how well the pre-trained multi-task models transfer to new downstream tasks. To this end, we construct a *downstream mix* consisting of three datasets with various degrees of similarity to the upstream mix. In particular, we select PASCALVOC-SP (Dwivedi et al., 2022b), PEPTIDES-STRUCT (Dwivedi et al., 2022b), and STARGAZERS (Rozemberczki et al., 2020; Morris et al., 2020).[4] Here, we measure performance for a varying number of fine-tuning steps to assess whether the pre-trained models are more sample-efficient than their single-task counterparts; see Figure 2 for the results. We observe that pre-training is generally beneficial in the regimes of the fewest optimization steps, although to a degree that depends on the target dataset and chosen backbone. We

---

[4]We detail the relation between "upstream" and "downstream" datasets in Appendix B.3.

observe strong in-domain transfer to PASCALVOC-SP and strong cross-domain transfer to STARGAZERS on both pre-trained models. The results on PEPTIDES-STRUCT are less pronounced. While we observe slight transfer for the pre-trained GT, the pre-trained MPNN shows negative transfer at 3K and 10K steps.

Overall, the above results suggest that MPNNs and GTs can learn general-purpose graph representations even when trained on data from different domains. These representations can often transfer effectively to in- and cross-domain tasks.

## 8. Conclusion

This paper highlights the need to rethink benchmarks and practices in graph learning. While GNNs have succeeded in many applications, current benchmarks often overlook real-world problems, focus too narrowly on specific data modalities, and lack consistent evaluation protocols or large-scale datasets for foundation models. We propose designing benchmarks that reflect real-world complexity, standardizing evaluations, and creating scalable datasets. These changes will help the graph learning community align with machine learning advancements and maintain impact and relevance.

## Acknowledgements

We thank Erik Müller for crafting the figures and Petar Veličković for feedback on the paper. AS, CM, and LM are partially funded by a DFG Emmy Noether grant (468502433) and RWTH Junior Principal Investigator Fellowship under Germany's Excellence Strategy. AS performed this work as part of the Helmholtz School for Data Science in Life, Earth and Energy (HDS-LEE) and received funding from the Helmholtz Association of German Research Centres. BF is funded by the Clarendon Scholarship. FF conducted this work while partly supported at the Technion by an Aly Kaufman and an Andrew and Erna Finci Viterbi Post-Doctoral Fellowship. MB is supported by EPSRC Turing AI World-Leading Research Fellowship No. EP/X040062/1 and EPSRC AI Hub on Mathematical Foundations of Intelligence: An "Erlangen Programme" for AI No. EP/Y028872/1.

## Impact Statement

This paper presents work whose goal is to advance the field of Machine Learning. There are many potential societal consequences of our work, none which we feel must be specifically highlighted here.

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

# A. Extended related work

GRAPHWORLD (Palowitch et al., 2022) offered a synthetic perspective on graph benchmarking by showing that existing datasets cover a relatively narrow distribution of possible graphs and tuning common MPNN architectures is not indicative of their performance in other, less common domains. To alleviate the distribution issue, GRAPHWORLD suggested generating synthetic graphs using stochastic block models (Karrer & Newman, 2011) with more diverse connectivity patterns and probing GNNs on the synthetic datasets. Unfortunately, the dataset did not receive significant attention and adoption in the graph learning community, partly due to the stated synthetic nature of the tasks.

In addition, Veličković et al. (2022) proposed CLRS, an algorithmic reasoning benchmark modeling the simulation of 30 classical algorithms as graph tasks such as node- or edge-level prediction and evaluates in a challenging size generalization setting. Algorithmic reasoning, in particular in the size generalization setting, receives some interest in the broader machine learning community (Zhou et al., 2024a;b; McLeish et al., 2024) and is arguably highly relevant to the study and advancement of the reasoning capabilities of neural networks in general. At the same time, algorithmic reasoning methods are typically benchmarked on synthetic tasks to study reasoning and learning capabilities in a controlled setting. There is no suitable replacement for high-quality, real-world benchmarks with direct downstream applications.

Furthermore, the benchmarking landscape for MPNNs remains constrained by the lack of large-scale and realistic graph datasets, particularly in domains like social networks, and commonly used datasets such as REDDIT (Hamilton et al., 2017) and FLICKR (Zeng et al., 2019) are often cited as representative of real-world social networks. However, these datasets fail to capture key characteristics of actual social networks, such as high-degree hubs and dense community structures, as their average node degree is significantly lower. This discrepancy makes them poor representatives for large, realistic graphs. Similarly, in social network-based datasets such as the Twitter retweet-induced subgraph dataset (Ribeiro et al., 2017), it is unclear whether the features and adjacency relationships of the sampled subgraph align with those of the full graph. Moreover, the structure of these subgraphs is constructed using a random walk-based crawler on the original graph. This sampling process further reduces the average node degree, making the dataset less representative of large-scale graphs, similar to REDDIT and FLICKR. We note here that the unavailability of real-world social network graph data is likely due to factors outside our field's control, e.g., privacy concerns and commercial relevance.

Another essential aspect of many real-world graphs is their inherently dynamic nature. That is, nodes, edges, and their features change over time. This aspect is often neglected in many datasets, including social networks. Recent efforts have introduced valuable benchmarking suites for learning on temporal graphs, e.g., the *Temporal Graph Benchmark* (TGB) (Huang et al., 2023; Gastinger et al., 2024a). Interestingly, researchers have found that overlooked baselines and simple heuristics can be particularly predictive on these temporal datasets and outperform more sophisticated temporal GNNs (Gastinger et al., 2024a;b). This puts in question the relevance of some of the proposed benchmarks and the significance of the progress made by the community.

# B. Additional experimental details

Here, we provide additional experimental details and results.

## B.1. Graphs not necessarily constructed in a meaningful way

For each model among GraphConv, GIN, and GAT, we tuned the learning rate in $\{10^{-3}, 5 \cdot 10^{-3}\}$, number of layers in $\{3, 5\}$, dropout in $\{0, 0.3\}$, hidden dimensions in $\{32, 64\}$, batch size in $\{16, 32\}$, early stopping with patience of 50 steps on the validation loss, and sum-pooling. We used ReLU activation and CrossEntropy loss.

We trained with seed 0 for each dataset over all the hyper-parameter configurations and selected the best-performing configuration on the validation set, according to the ROC-AUC scores. We then trained each model with its selected configuration with seeds 1 and 2. Finally, we report the mean and standard deviation of the ROC-AUC scores over the test set over these 3 seeds.

We consider the CGP propagation scheme from Wilson et al. (2024), where for each model, we utilize the Cayley graph in each layer and do not consider the original graph at all. For the DeepSet evaluation, we used the same architecture of GraphConv and fed it with empty graphs.

*Table 5.* Best hyper-parameters in the format num. of layers, width, batch size, learning rate, dropout.

| Model | OGBG-MOLHIV | OGBG-MOLBBBP | OGBG-MOLBACE |
|---|---|---|---|
| GraphConv Orig. | $3, 64, 32, 5e{-}4, 0$ | $5, 32, 16, 1e{-}4, 0$ | $5, 32, 32, 5e{-}4, 0$ |
| GraphConv Empty | $3, 64, 32, 1e{-}4, 0$ | $3, 64, 32, 5e{-}4, 0$ | $3, 32, 16, 5e{-}4, 0.3$ |
| GraphConv Cayley | $3, 64, 32, 5e{-}4, 0.3$ | $3, 64, 32, 5e{-}4, 0$ | $5, 64, 32, 5e{-}4, 0$ |
| GIN Orig. | $5, 32, 32, 1e{-}4, 0$ | $3, 64, 16, 1e{-}4, 0$ | $3, 32, 32, 1e{-}4, 0.3$ |
| GIN Empty | $3, 64, 32, 5e{-}4, 0.3$ | $3, 32, 32, 5e{-}4, 0$ | $3, 32, 32, 1e{-}4, 0.3$ |
| GIN Cayley | $5, 32, 32, 1e{-}4, 0$ | $5, 64, 64, 5e{-}4, 0$ | $5, 64, 16, 1e{-}4, 0$ |
| GAT Orig. | $5, 64, 32, 5e{-}4, 0$ | $3, 64, 16, 5e{-}4, 0$ | $5, 32, 32, 1e{-}4, 0.3$ |
| GAT Empty | $3, 32, 32, 1e{-}4, 0$ | $3, 32, 32, 5e{-}4, 0$ | $5, 32, 32, 1e{-}4, 0.3$ |
| GAT Cayley | $3, 64, 32, 1e{-}4, 0$ | $3, 64, 32, 5e{-}4, 0$ | $5, 64, 16, 1e{-}4, 0$ |

## B.2. Reassessing simple baselines on PCQM4Mv2

We make minor changes to the layer configuration by using SiLU activation (Hendrycks & Gimpel, 2016) instead of ReLU for improved gradient flow. We also replace the BatchNorm with LayerNorm and apply it *after* the skip connection, similar to a standard transformer encoder layer. In each GINE layer, we use a 2-layer MLP as an update function with a hidden dimension of 1024 (double the embedding dimension (512)). After the final GINE layer, we apply sum-pooling followed by a 3-layer MLP, outputting a graph-level prediction.

We set the number of GINE layers to 20 and the latent embedding dimension to 512. We found larger models to be overfitting in preliminary experiments, so we fixed this model configuration with approximately 20 million trainable parameters. This is comparable to the model sizes used in the graph transformer literature and about five times larger than the GINE model used to obtain the original results. Note that despite a model depth of 20 layers, we observed no performance degradation due to over-smoothing, trivially mitigated by following basic deep learning practices such as skip-connections and deep MLPs as update functions. It is known, of course, that these practices also prevent smoothing phenomena in transformers (Dong et al., 2021), and the same holds for MPNNs.

We train with the L1-loss for one million gradient descent steps using a batch size of 512 with the Adam optimizer. The learning rate warms up linearly for the first $10^4$ steps and follows a cosine decay schedule for the remainder of training towards a minimum rate of $10^{-6}$. No gradient clipping is used. We tune the remaining hyper-parameters through a grid search. Specifically, we tune the learning rate in $\{2{\cdot}10^{-4}, 1{\cdot}10^{-4}, 5{\cdot}10^{-5}\}$, the dropout rate in $\{0, 1{\cdot}10^{-1}, 2{\cdot}10^{-1}\}$ and the weight decay in $\{0, 1{\cdot}10^{-1}\}$. Tuning is done *with* RWSE features, and we reuse the same configuration for GINE without RWSE. Since the original validation split of PCQM4Mv2 is used to compare models in the literature, we create a separate holdout set by sampling 10K graphs uniformly at random from the training data and use this set for model selection and hyperparameter tuning. Each training run uses a single Nvidia H100 GPU and lasts approximately 8 hours. In total, hyperparameter tuning consumed less than 200 H100 hours of computing. The final hyper-parameters are provided in Table 6. For the final results reported in Table 2, we average the performance over three runs with different random seeds and also provide the corresponding standard deviation, which is relatively low.

*Table 6.* Hyper-parameters used for our evaluation of GINE on PCQM4Mv2.

| Hyperparameter | Value |
|---|---|
| learning rate | $2{\cdot}10^{-4}$ |
| weight decay | 0.1 |
| batch size | 512 |
| training steps | $10^6$ |
| warmup steps | $10^4$ |
| number of layers | 20 |
| embedding dimension | 512 |
| dropout | 0.1 |
| RWSE dimension | 20 |

## B.3. Multi-task pre-training with encoder-processor-decoder

Here, we outline details for our experiments with the encoder-processor-decoder setup.

### B.3.1. MODEL ARCHITECTURES

We consider an encoder-processor-decoder setup and two different processors, an MPNN with GINE (Hu et al., 2020b) layers and a graph transformer derived from Graphormer (Ying et al., 2021). As a common practice in the case of transformer architectures on graphs, we also experiment with injecting node-wise structural encodings, particularly RWSEs (Dwivedi et al., 2022a). In what follows, we detail the encoder-processor-decoder setup.

**Encoder**  Using task-specific encoders, we embed node and edge features into a standard embedding dimension $d \in \mathbb{N}^+$ for both architectures. If no node or edge features are available, we use learnable vectors that we train jointly with the architecture. Following standard practice in (graph) transformer encoders (Ying et al., 2021), we add a [cls] token from which we read out graph-level representations.

**Processor**  Subsequently, a processor network computes node- and graph-level representations from the embedded node, edge features, and graph structure. Given a graph $G$, both the MPNN and graph transformer update node representations $\mathbf{X} \in \mathbb{R}^{n \times d}$ at each layer as

$$\mathbf{X}' \leftarrow \mathbf{X} + \phi(\mathsf{LayerNorm}(\mathbf{X}), G),$$
$$\mathbf{X}'' \leftarrow \mathbf{X}' + \mathsf{MLP}(\mathsf{LayerNorm}(\mathbf{X}')),$$

where $\mathsf{MLP}$ is a two-layer MLP with GELU non-linearity (Hendrycks & Gimpel, 2016) and $\phi(\cdot, G)$ is either a graph convolution or attention, conditioned on $G$. Graph convolution is implicitly conditioned via message-passing over the local neighborhood.

In the case of attention, we add a graph-aware attention bias to the unnormalized attention matrix. Concretely, the graph transformer layer computes full multi-head scaled-dot-product attention over node-level tokens with a soft-attention bias computed from the edge features of the graph. The attention bias is a tensor $\mathbf{B} \in \mathbb{R}^{L \times L \times h}$, where $L \in \mathbb{N}^+$ is the number of tokens and $h \in \mathbb{N}^+$ is the number of attention heads. In particular, we compute a separate attention bias for each attention head. For a graph with $n$ nodes, we set $L := n + 1$ (accounting for the [cls] tokens). For simplicity, we write $i \in \mathbb{N}^+$ to indicate the $i$th node in an arbitrary but fixed node ordering. We refer to the [cls] tokens as node $n + 1$. Further, only for the graph transformer, we use a maximum context size of 8192 and remove additional nodes that exceed this size. We then compute the attention bias $\mathbf{B}$ such that for all edges $(i, j)$,

$$\mathbf{B}_{ij} := \mathbf{W} \cdot \mathbf{e}_{ij},$$

where $\mathbf{e}_{ij} \in \mathbb{R}^d$ is the edge feature of $(i, j)$ and $\mathbf{W} \in \mathbb{R}^{d \times h}$ is a learnable weight matrix. Again, we omit bias terms for clarity. If no edge exists between nodes $i$ and $j$, we set $\mathbf{B}_{ij}$ to all-zeros. For the [cls] token, we use learnable vectors $\mathbf{e}_{\mathrm{in}}, \mathbf{e}_{\mathrm{out}} \in \mathbb{R}^d$ as the attention bias for in- and out-coming edges, respectively, i.e., we set

$$\mathbf{B}_{(n+1)j} := \mathbf{e}_{\mathrm{in}},$$
$$\mathbf{B}_{i(n+1)} := \mathbf{e}_{\mathrm{out}}.$$

Finally, we add $\mathbf{B}$ as a soft bias to the unnormalized attention matrix, that is, before applying softmax.

**Decoder**  Lastly, we apply a decoder network that makes task-specific predictions. In our experiments, we used the same MLP layout for all decoders. In particular, given a representation vector $\mathbf{x} \in \mathbb{R}^d$, we define our decoder MLP as

$$\mathbf{W}_2 \mathsf{LayerNorm}(\mathsf{GELU}(\mathbf{W}_1 \mathbf{x})),$$

where $\mathbf{W}_1 \in \mathbb{R}^{d \times d}$, $\mathbf{W}_2 \in \mathbb{R}^{d \times o}$ are learnable weight matrices, with $o \in \mathbb{N}^+$ the task-specific output dimensions (e.g., the number of classes in a classification task) and a GELU non-linearity (Hendrycks & Gimpel, 2016). We omit bias terms for clarity.

### B.3.2. MULTI-TASK PRE-TRAINING

Here, we outline details on the multi-task pre-training.

**Training loop and optimization parameters**  We perform multi-task pre-training by using data loaders for all tasks and accumulating gradients from each task at each iteration of the training loop, effectively simulating a "heterogeneous batch" of data from all available tasks. We train on bfloat16 with clipped gradients and a cosine learning rate scheduler.

*Table 7.* Datasets in the pretraining mix.

| Dataset | Domain | Task | Avg. # nodes | Avg. # edges |
|---|---|---|---|---|
| COCO-SP | Vision (Super Pixels) | Semantic Segmentation | 476.88 | 2 693.67 |
| MALNETTINY | Cybersecurity (Function Calls Graphs) | Malware Detection | 1 410.3 | 2 859.9 |
| PCQM4Mv2 | Chemistry (Small 2D Molecules) | HOMO-LUMO Gap Prediction | 14.1 | 14.6 |

**Pretraining mix**  As already mentioned in the main text, the pretraining mix is formed by the datasets described in Table 7, which differ in domain, task, and structural properties.

- COCO-SP (Dwivedi et al., 2022b) is a dataset of sparse, medium-sized graphs encoding images at the super-pixel level. Nodes, i.e., super-pixels, are attributed with pixel value statistics and center-of-mass coordinates. The task is to predict, for each superpixel, a semantic segmentation label.
- PCQM4Mv2 (Hu et al., 2021a) comprises many small molecular graphs for which the task is to predict the HOMO-LUMO energy gap. Interestingly, only accessing 2D molecular information is practically relevant in this setting, as calculating 3D structures requires expensive DFT-based geometry optimization.[5]
- MALNET-TINY(Freitas et al., 2021) includes a relatively small number of larger graphs encoding function calls, with the task being to predict their association with malicious code execution. These graphs are entirely unattributed.

**Hyperparameter tuning**  We sweep the learning rate over $\{4 \cdot 10^{-5}, 7 \cdot 10^{-5}, \ldots, 1 \cdot 10^{-3}\}$ for graph transformers and $\{4 \cdot 10^{-5}, 7 \cdot 10^{-5}, \ldots, 1 \cdot 10^{-2}\}$ for GINE and train for 100K gradient steps. We pick the pre-trained checkpoint based on the best overall validation loss, which we compute as the sum of all three task losses.

### B.3.3. SINGLE-TASK FINE-TUNING

Here, we outline details on the single-task fine-tuning.

**Architectural details**  Across all finetuning experiments, the prediction heads (i.e., the decoders) are initialized and trained from scratch, while the (pre-trained) processors are kept frozen. If a downstream task shares the same node and/or edge features with a pretraining dataset, we reuse the corresponding (pre-trained) encoders, which are also frozen during finetuning. Otherwise, a new encoder is initialized and trained from scratch. Note that downstream datasets with featureless nodes share identical (pre-trained) encoders; see, e.g., STARGAZERS below.

In all cases, we run a standard single-task finetuning on bfloat16 with clipped gradients and a cosine learning rate scheduler.

*Table 8.* Datasets considered for downstream applications.

| Dataset | Domain | Task | Avg. # nodes | Avg. # edges |
|---|---|---|---|---|
| PASCALVOC-SP | Vision (Super Pixels) | Semantic Segmentation | 479.40 | 2 710.5 |
| PEPTIDES-STRUCT | Chemistry (Peptides) | 3D-Structure Property Prediction | 150.9 | 307.3 |
| STARGAZERS | Github Communities | Social Network Classification | 113.79 | 234.64 |

**Downstream (finetuning) datasets**  The datasets considered as downstream applications for our finetuning experiments are enlisted and concisely described in Table 8. Again, they vary widely in domains, tasks, and structural features while encompassing various levels of similarity to the datasets in the pretraining mix. In particular:

- PASCALVOC-SP (Dwivedi et al., 2022b) is aligned with COCO-SP in most aspects: domain, task and structure. Here, we can reuse the pre-trained encoder of COCO-SP.

---

[5]See https://ogb.stanford.edu/docs/lsc/pcqm4mv2/.

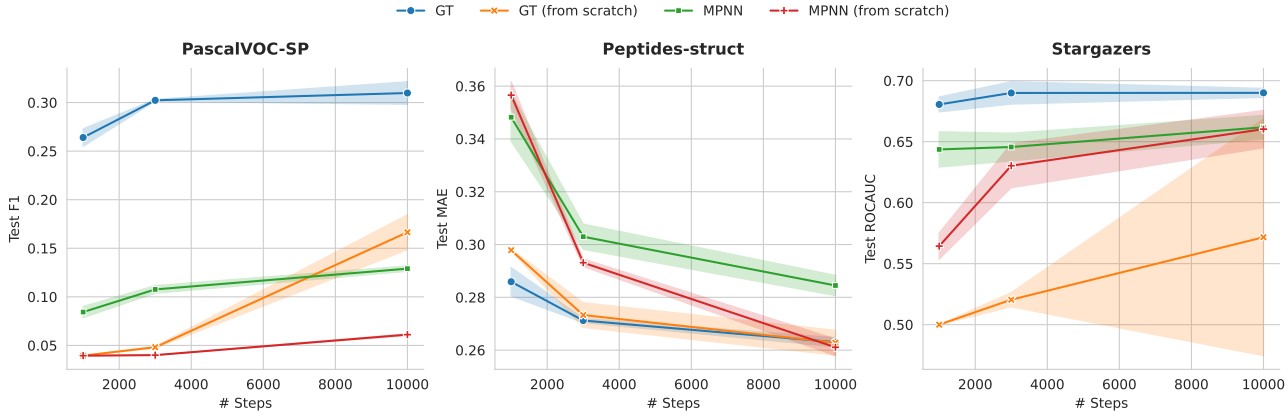

*Figure 2.* **Highlights** from our fine-tuning results on three downstream datasets with varying numbers of fine-tuning steps and varying degrees of similarity to the upstream mix.

- PEPTIDES-STRUCT (Dwivedi et al., 2022b) comprises molecular graphs and belongs to the same broad chemical domain of PCQM4MV2, from which our downstream model reuses the feature encoder. However, these molecular graphs are distinct (they represent chains of amino acids) and structurally different (they are larger and more elongated, with higher diameter values). The task is also different in that it pertains to predicting 3D structural features rather than quantum properties.

- STARGAZERS (Rozemberczki et al., 2020; Morris et al., 2020) comprises social networks formed by GitHub developers who have starred at least 10 repositories, connected by 'following' relations. The task is to classify these social networks as belonging to either Web or Machine Learning developers. This dataset is entirely different in task and domain from any datasets considered in the pretraining mix. No node or edge features are available, so our model reuses the same node encoder pre-trained on the featureless MalNetTiny.

**Training setup** In this setting, we are interested in measuring the sample efficiency of our models, aiming to study if and when pre-training is beneficial. Accordingly, we train for 1K, 3K, and 10K steps, setting the batch size to correspond to roughly 2, 10, and 30 epochs. The same setting is employed for reference single-task baselines, trained from scratch on the same amount of data.

**Hyperparameter tuning** The learning rate is the only tuned hyper-parameter; we sweep over 3 orders of magnitude in $\{1 \cdot 10^{-5}, 4 \cdot 10^{-5}, 7 \cdot 10^{-5}, \dots, 1 \cdot 10^{-2}\}$ for each fine-tuning regime. The single-task baselines are always sized in a way to total the same number of parameters of their multi-task pre-trained counterparts.

**Additional results** In addition to Figure 2, we provide additional fine-tuning results in Figure 3, where we compare the GNN with additional RWSE, as well as the graph transformer without additional structural encodings.

### B.4. Variance of results reported on ENZYMES

In Section 4, we discussed problematic practices prevalent in experimental evaluations of GNNs. A common problem is using small, high-variance datasets without an established evaluation protocol. For some datasets, the numbers reported throughout the literature vary substantially, resulting in an inconsistent and confusing representation of model performance. Here, we illustrate this problem for the commonly used ENZYMES dataset (Morris et al., 2020) as an example of how extreme reported performance measurements vary.

In Figure 4, we plot the reported test accuracy of different graph learning publications on the ENZYMES dataset against the year of publication. We include results from various lines of work, such as graph kernels (Shervashidze et al., 2011), GNN benchmarking (Dwivedi et al., 2020), graph pooling (Ying et al., 2018; Feng & Weber, 2024), and graph rewiring (Karhadkar et al., 2023; Barbero et al., 2024). The results reveal an interesting trend. Older kernel-based methods, such as the WL kernel, achieved a baseline accuracy of around 52%. Initial evaluations of MPNN models outperformed this

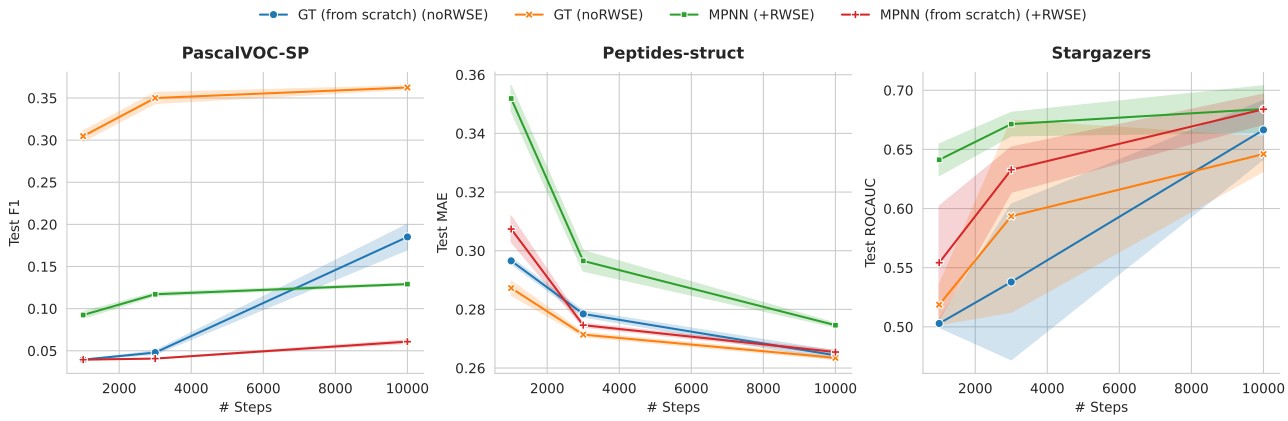

*Figure 3.* Additional downstream results for MPNN+RWSE and graph transformers without additional structural encodings. Note that the GT is still graph-aware due to the soft attention bias.

baseline, often achieving over 60%, even with simple GCN-based models. However, newer publications from 2023 and 2024 often fall short of these results by a significant margin, sometimes reporting less than 30% classification accuracy, even when evaluating similar GCN-based architectures. In other words, the results reported for the same base architecture can vary by a factor of two across publications.

There are several causes for this extreme variance. First of all, there is no consistent evaluation setup. While older publications typically used stratified 10-fold cross-validation as an evaluation protocol, newer results are often based on repeated random 80/10/10 splits, which are prone to be noisier. This difference explains the performance variance to some degree but does not account for the sharp drop in the reported accuracy of more recent publications. Instead, many recent works seem to run experiments with suboptimal hyperparameter choices, resulting in a significant loss in performance for the compared models. For example, Barbero et al. (2024) configure the training to only last 100 epochs, which is too short to allow for model convergence on a dataset as small as ENZYMES.

While each publication is internally consistent in that it applies the same experimental setup to the methods it compares, one can argue that this is insufficient when measurements vary drastically over time. The lack of standardization risks conflating methodological improvements with artifacts of experimental design. Ensuring cross-study consistency—through adherence to shared protocols and rigorous benchmarking on more suitable datasets—is critical to fostering trust in reported results and enabling clear advancements in graph learning research.

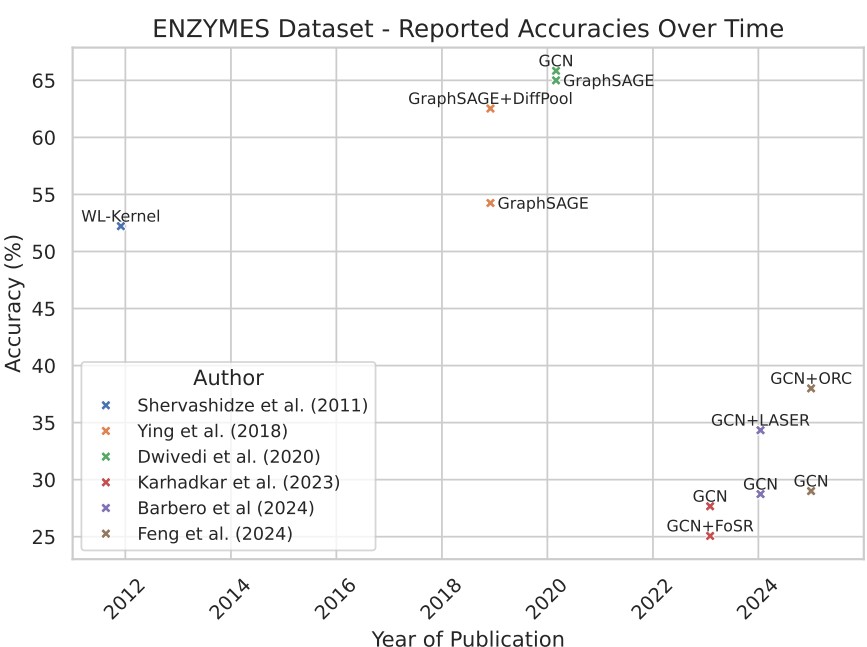

*Figure 4.* Test accuracy reported on the ENZYMES dataset over the past twelve years in various publications: Shervashidze et al. (2011); Ying et al. (2018); Dwivedi et al. (2020); Karhadkar et al. (2023); Barbero et al. (2024); Feng & Weber (2024)

