# OpenReview forum: "Position: Graph Learning Will Lose Relevance Due To Poor Benchmarks"
_ICML.cc/2025/Position_Paper_Track — ICML 2025 Position Paper Track poster_

### Official Review · Reviewer_zadS · 2025-02-22

**Significance:** 2
**Argument Clarity:** 3
**Rating:** 3
**Confidence:** 4

**Questions:**

1. How does “Relevance” in the title mean? I am not sure graph learning will lose relevance to what?
2. In the highlighted sentence at the end of the first page, how does “design of current datasets, the investigated data modalities, and current benchmarking practices” correspond to the following sections?
3. The paper suggests that graph learning benchmarks should focus on large scale and real-world graph datasets. Why the empirical study part does not create such datasets and conduct experiments to serve as an example?
4. In your opinion, what is relation between GNNs and graph transformer? How can they contribute to graph foundation model, respectively?

**Discussion Potential:**

3

**Paper Summary:**

This position paper argues that the field of graph learning must reassess its benchmarking practices in order to maintain its impact and relevance. To support this perspective, the paper highlights three key shortcomings in existing research: the lack of support for real-world applications, the use of irrelevant graph construction methods, and the prevalence of poor benchmarking practices. The paper also advocates for the development of a graph foundation model based on an encoder-processor-decoder framework. In addition, empirical evidences are provided to reinforce the proposed arguments.

**Position:**

Yes

**Position In Title:**

Yes

**Related Work:**

3

**Strengths And Weaknesses:**

Strength:
1. The paper focus on the problem of poor benchmarking of existing graph learning works and provides evidences to support the arguments.
2. The empirical evidence part reflects the common problems of existing graph learning works.

Weakness:
1. In section 2, 3 and 4, the paper supports its viewpoints through providing examples. However, the paper does not provide enough evidence to show that the phenomenon in examples widely exists. It is more persuasive statistical data (collection) of existing graph learning benchmarks to support the arguments.
2. Link prediction is an important branch of graph learning, but this paper rarely mentions the datasets or applications related to link prediction tasks.
3. In section 2, 3 and 4, the paper all use examples of tasks related to molecular graphs to verify its viewpoints, which implies that arguments in this paper maybe one-sided.
4. Section 6 seems not necessary to support the viewpoint of this paper. In section 6, the relations between geometric deep learning and graph learning are also not introduced in detail.

**Support:**

2

---

> ### Author Rebuttal · Authors · 2025-03-27
>
> > In section 2, 3 and 4, the paper supports its viewpoints through providing examples. However, the paper does not provide enough evidence [...]
>
> > Link prediction is an important branch of graph learning, but [...]
>
> > In section 2, 3 and 4, the paper all use examples of tasks related to molecular graphs [...]
>
> > Section 6 seems not necessary to support the viewpoint of this paper. [...]
>
> > How does “Relevance” in the title mean? [...]
>
> > In the highlighted sentence at the end of the first page, how does “design of current datasets, the investigated data modalities, and current benchmarking practices” correspond to the following sections?
>
> > The paper suggests that graph learning benchmarks should focus on large scale and real-world graph datasets. Why the empirical study part does not create such datasets [...]
>
> > In your opinion, what is relation between GNNs and graph transformer? How can they contribute to graph foundation model, respectively?
>
> **Statistical Evidence for Benchmarking Issues**
>
>  Our empirical study (Section 7) highlights benchmarking inconsistencies, but we agree that a broader statistical analysis would strengthen our position. Our next revision will expand the discussion with more datasets, reinforcing the prevalence of these issues. However, our goal is not just to list flawed benchmarks but to highlight systemic problems in the graph ML benchmarking culture, some of which may be more difficult to discuss quantitatively.
>
> **Coverage of Link Prediction Benchmarks**
>
>  The reviewer is correct that link prediction is essential. While we focus on graph- and node-level tasks, our critique applies to link prediction. We will expand our discussion to include benchmarks from knowledge graphs, social networks, and recommender systems (e.g., RelBench), highlighting challenges like biased negative sampling, dataset leakage, and inconsistent evaluation.
>
> **Overemphasis on Molecular Graphs**
>
>  Molecular datasets are widely studied and illustrate benchmarking flaws well., On one hand, this reveals a problematic overfocus of the graph learning community on this specific domain, and contributes to strengthening our position. On the other, we acknowledge the concern that our focus may appear one-sided. As mentioned above, we will incorporate examples from social networks, recommendation systems, and combinatorial optimization to show these issues are pervasive.
>
>
> **Relevance of Geometric Deep Learning (GDL) Discussion**
>
>  Section 6 contrasts graph ML benchmarking struggles with GDL’s success, aligning with real-world applications. We will make this connection more explicit by discussing key benchmarking differences.
>
>
> **Clarifying “Relevance” in the Title**
>
>  By "losing relevance," we mean that poor benchmarking practices risk disconnecting graph learning from impactful real-world applications. This includes: Benchmarks failing to represent real-world tasks, leading to poor model generalization in practical, industrial applications. Stagnation as research focuses on flawed benchmarks instead of meaningful problems. We will clarify this in the introduction.
>
>
> **Connection Between Key Issues and Paper Structure**
>  The reviewer asks how the dataset design, data modalities, and benchmarking practices connect to the paper. We clarify:
> - Section 2: How benchmarks fail to capture real-world problems (dataset design).
> - Section 3: Artificial graph construction without meaningful task connection (data modalities).
> - Section 4: Inconsistent evaluation and benchmarking (methodology).
>
>  We will refine transitions to tie these themes to their respective sections explicitly.
>
>
> **Why Not Propose a New Benchmark?**
>
>  The reviewer suggests creating a large-scale benchmark. While this would be valuable, our paper critiques current practices and outlines principles for future benchmarks rather than introducing one. Developing a high-quality benchmark requires community effort –while we anticipate we are already actively working on this, we also believe this goes beyond the scope of a position paper.
>
>
> **GNNs vs. Graph Transformers in Foundation Models**
>  The reviewer asks about the roles of GNNs and graph transformers. We clarify:
> - GNNs leverage explicit graph structure via message passing, making them ideal for tasks with strong relational inductive biases.
> - Using attention mechanisms and learned structure, graph transformers are well-suited for heterogeneous graphs, large-scale pretraining, and multi-modal tasks.
> A hybrid approach, combining GNN-style local processing with global attention, is likely the most promising direction, and initial progress has been made in this direction. We hope our call for a better benchmarking culture and infrastructure will catalyze endeavours around this research question.
>
>  We will expand on this discussion in the manuscript revision.
>
> **Please consider updating your score if you are satisfied with our answer. We are happy to answer any remaining questions.**

---

> > ### Comment · Reviewer_zadS · 2025-04-02
> >
> > I appreciate the authors' detailed response, which well addresses my questions. After reviewing the comments from other reviewers and the authors' reply, I decide to maintain my current score.

---

> > > ### Author Response · Authors · 2025-04-02
> > >
> > > Thank you for your prompt response. Please let us know what could potentially prevent you from increasing your score.

---

### Official Review · Reviewer_wSEM · 2025-03-09

**Significance:** 4
**Argument Clarity:** 3
**Rating:** 4
**Confidence:** 4

**Questions:**

See Weaknesses and Questions

**Discussion Potential:**

4

**Paper Summary:**

In this paper, the authors present perspectives regarding benchmarking graph machine learning. Specifically, the authors argue that existing benchmarks focus on narrow domains (e.g., citation graphs and 2-D molecules), have poor representations (e.g., inadequate abstractions and misaligned use cases), and fragmented evaluations (e.g., incentivizing over-fittings). Therefore, the authors advocate for new benchmarks, including expanding the domains (e.g., combinatorial optimization, relation database, chip design), clarifying why to use graphs, rigorous evaluation protocols (e.g., standard datasets split and hyper-parameters), and calling for collaboration with domain experts.

**Position:**

Yes

**Position In Title:**

Yes

**Related Work:**

3

**Strengths And Weaknesses:**

Strengths:
1. The authors have identified a critical problem in graph machine learning. Personally, I think many related researchers sense this problem but do not take the time to delve deeply into it.
2. The authors have provided empirical results to support some claimed disadvantages of the current benchmark.
3. The authors have proposed many remedies for the identified problems.

Weaknesses and Questions:
1.	Though I acknowledge the benchmark problem of graph machine learning cannot be solved in one position paper, I feel the paper could be more meaningful if more discussions could be provided on how to push forward the benchmark problem truly. Some detailed points are as follows:
(a) I wonder whether the authors have considered how to advocate the usage of a newly proposed benchmark. One bottleneck, as mentioned in the paper, is that there exists an inherent difficulty in scaling to large graphs. This is somewhat different from the NLP community, where developing LLMs can be incredibly expensive, but testing LLMs will be relatively easy. On the contrary, testing graph machine learning models on large benchmarks may not be feasible for researchers without access to sufficient computational resources.
(b) Another issue I feel worthy of discussing is that, considering many existing graph benchmarks (or called datasets) are available, e.g., in each year’s NeurIPS benchmarking track, it may be more feasible and economical to integrate these separate resources into a benchmark, rather than developing a new benchmark from scratch.

2.	Regarding the graph foundation model aspect, the authors focus more on classical GNN-based approaches. On the other hand, there is a recent trend of using or combining LLMs for graph machine learning, particularly for text-attributed graphs. As a result, many graph LLM benchmarks have also recently been proposed. I wonder what are the authors‘ opinions in using these benchmarks as general graph machine learning benchmarks.

3.	Some presentations could be improved, e.g., Fig 1 could be better illustrated.

**Support:**

4

---

> ### Author Rebuttal · Authors · 2025-03-27
>
> > 1. Though I acknowledge the benchmark problem of graph machine learning cannot be solved in one position paper, I feel the paper could be more meaningful if more discussions could be provided on how to push forward the benchmark problem truly. Some detailed points are as follows:
>
> > (a) I wonder whether the authors have considered how to advocate the usage of a newly proposed benchmark. One bottleneck, as mentioned in the paper, is that there exists an inherent difficulty in scaling to large graphs. This is somewhat different from the NLP community, where developing LLMs can be incredibly expensive, but testing LLMs will be relatively easy. On the contrary, testing graph machine learning models on large benchmarks may not be feasible for researchers without access to sufficient computational resources.
>
> > (b) Another issue I feel worthy of discussing is that, considering many existing graph benchmarks (or called datasets) are available, e.g., in each year’s NeurIPS benchmarking track, it may be more feasible and economical to integrate these separate resources into a benchmark, rather than developing a new benchmark from scratch.
>
> > Regarding the graph foundation model aspect, the authors focus more on classical GNN-based approaches. On the other hand, there is a recent trend of using or combining LLMs for graph machine learning, particularly for text-attributed graphs. As a result, many graph LLM benchmarks have also recently been proposed. I wonder what are the authors‘ opinions in using these benchmarks as general graph machine learning benchmarks.
>
> > Some presentations could be improved, e.g., Fig 1 could be better illustrated.
>
>
> We sincerely appreciate the reviewer’s thoughtful comments and valuable suggestions. Below, we address each concern in detail:
>
> **1. How to Advocate the Usage of New Benchmarks**
>
> We agree that merely proposing a new benchmark is insufficient—adoption is a key challenge. To address this:
> We emphasize that meaningful benchmarks must be designed with usability and scalability in mind, considering computational constraints. For instance, modular datasets with subsets of varying sizes can enable both resource-constrained researchers and large-scale teams to evaluate models fairly. We acknowledge the reviewer’s point that large-scale benchmarking in graphs can be computationally expensive, and we refine our position to advocate for scalable benchmarks that accommodate different computational resources.
> We advocate for community-driven efforts, including leaderboards, standard splits, and rigorous evaluation protocols. Open Graph Benchmark (OGB) has demonstrated how centralization can drive adoption, and we encourage similar initiatives.
>
> We highlight the importance of industry-academic collaboration, as seen in other ML subfields, to facilitate access to high-quality datasets that may otherwise be unavailable.
>
> **2. Integrating Existing Benchmarks vs. Creating New Ones**
>
> We fully agree that integrating and improving existing resources is often more feasible than developing new benchmarks from scratch. Our argument is not that we just need entirely new datasets, but rather that existing benchmarks require restructuring to align with real-world applications and fair evaluation standards. Some specific points:
> Many current datasets suffer from inconsistent evaluation protocols, poor task formulation, or lack of real-world grounding (as detailed in Sections 2 and 4).
>
> A potential approach is redefining and standardizing evaluation procedures across existing datasets rather than accumulating more datasets. The field would benefit from benchmark aggregation platforms that incorporate multiple datasets into a cohesive benchmarking framework with well-defined evaluation metrics.
>
> Thus, while we discuss the need for improved benchmarks, we do not necessarily advocate for redundant new datasets but rather for a principled restructuring of benchmarking practices.
>
> **3. The Role of Graph LLM Benchmarks**
>
> The reviewer raises an excellent point regarding LLM-based graph learning. We acknowledge that LLMs have introduced a new paradigm for graph-based tasks, particularly for text-attributed and knowledge graphs. However:
> Our focus remains on evaluating general graph ML benchmarks, including but not limited to text-attributed graphs. While LLM-based methods are promising, they are not a substitute for structured graph-based reasoning, especially in domains like combinatorial optimization or molecular modeling. LLM-based graph benchmarks can complement traditional ones, and we support their inclusion in broader benchmarking discussions.
>
> In our revision, we will clarify that our position does not exclude emerging trends like graph LLMs but highlights the need for holistic benchmarking that spans various graph ML paradigms.
>
> **4. Presentation Improvements**
>
> We appreciate the reviewer’s feedback on Figure 1 and will revise it for better clarity and illustration of key ideas.

---

> > ### Comment · Reviewer_wSEM · 2025-04-03
> >
> > I have read the rebuttal and encourage the authors to revise the manuscript based on these comments. I remain positive about the paper.

---

### Official Review · Reviewer_AfXm · 2025-03-09

**Significance:** 3
**Argument Clarity:** 2
**Rating:** 2
**Confidence:** 4

**Questions:**

* It is a bit ambiguous about the way to define the so called transformative real-world applications. Do you mean we need good datasets for each of these kinds of applications?
* Many applications you mention in the section 2 already have extensive studies such as recommender system and social network. I feel like other ones are just tasks that GNN models are utilized recently. The existence of comprehensive benchmark is naturally lacking.

**Discussion Potential:**

4

**Paper Summary:**

The paper argues that graph learning will lose relevance due to poor benchmarks. Specifically, current benchmark lacks transformative real-world applications. The authors propose to re-design datasets, investigate data modality and benchmark pipeline.

**Position:**

Yes

**Position In Title:**

Yes

**Related Work:**

3

**Strengths And Weaknesses:**

* Strength
  * The paper proposes a long-standing statement toward the foundation of graph machine learning, which is relatively less discussed recently.
   * Section 4 is specifically important and relevant in the field of graph learning.
  * Empirical evidence is also important to make a proper position.


* Weakness
   * It feels like the scope is very ambitious in the title; while the content does not support the claim too much since the applications mentioned in the paper are quite limited.
   * Section 3 is weak since it is relatively easy to check the effectiveness of structure. A MLP or deep-set can be trained to validate whether structure can help. This comparison is straightforward and should be something too critical.
   * Section 5 seems a bit disconnected to the position.
   * It feels like the position paper leans toward to molecule generation. I think the title and the main claim should somehow be adjusted to this scope. The discussion on other perspectives is relatively narrow.

**Support:**

2

---

> ### Author Rebuttal · Authors · 2025-03-27
>
> > - It feels like the scope is very ambitious in the title; while the content does not support the claim too much since the applications mentioned in the paper are quite limited.
>
> > - Section 3 is weak since it is relatively easy to check the effectiveness of structure. A MLP or deep-set can be trained to validate whether structure can help. This comparison is straightforward and should be something too critical.
>
> > - Section 5 seems a bit disconnected to the position.
>
> > - It feels like the position paper leans toward to molecule generation. I think the title and the main claim should somehow be adjusted to this scope. The discussion on other perspectives is relatively narrow.
>
> > - The definition of transformative real-world applications is a bit ambiguous. Do you mean we need suitable datasets for each of these kinds of applications?
>
> > - Many applications you mention in the section 2 already have extensive studies such as recommender system and social network. [...]
>
> We appreciate the reviewer’s detailed feedback and the opportunity to clarify the scope and contributions of our position paper.
>
> Below, we address the key points raised in the review:
>
> **Scope and Ambition of the Paper**
>
>  While our title highlights a broad challenge in graph learning, our position is not merely speculative but grounded in concrete examples of benchmarking failures and proposed remedies. The applications we discuss—combinatorial optimization, relational databases, and chip design—are intentionally chosen to illustrate diverse domains where improved benchmarking practices could have a transformative impact. While we acknowledge that our empirical examples are necessarily limited, our broader argument applies across various applications.
>
>
> **Section 3: The Role of Unstructured Baselines**
>
>  The reviewer suggests that validating the structure's usefulness is trivial by comparing against MLPs or DeepSets. However, our argument is not merely about whether structure helps but when and how graph structure is meaningfully utilized. Prior work, including Bechler-Speicher et al. (2024), has demonstrated cases where GNNs overfit to spurious structural correlations, leading to misleading conclusions about their advantages. Our discussion highlights that benchmarking should rigorously evaluate whether relational inductive biases genuinely contribute to generalization.
>
>
> **Relevance of Section 5 to Our Position**
>
>  The reviewer finds Section 5 somewhat disconnected, but this is a natural consequence of our position: inadequate benchmarks hinder the development of foundation models in graph learning. Unlike NLP or vision, where benchmark-driven advancements have led to general-purpose foundation models, graph learning remains fragmented due to its problematic benchmarking culture. We explicitly call for a shift toward unified, large-scale datasets to address this gap.
>
> **Focus on Molecular Generation**
>
>  While molecular datasets have been widely used in benchmarking, we explicitly critique their overrepresentation and advocate for moving beyond them to areas such as combinatorial optimization and social networks. Our argument is not to reinforce molecule generation as the central domain but rather to challenge the narrow scope of existing benchmarks and propose more diverse, problem-driven alternatives.
>
>
> **Definition of Transformative Real-World Applications**
>
>  We agree that the definition of "transformative" applications could be more precise, and we refine our position as follows: we advocate for benchmarks that (1) represent problems with real-world impact, (2) justify the use of graph structure, and (3) include rigorous baselines that assess the necessity of graph representations. This includes applications where graphs arise naturally (e.g., combinatorial optimization, power networks) rather than artificially imposed structures.
>
>
> **Prevalence of Graph ML in Certain Applications**
>
>  The reviewer notes that some applications we mention (e.g., recommender systems, social networks) already have extensive studies. We acknowledge this. However, our position is not that these areas are underexplored; instead, our critique is that the benchmarks often fail to provide meaningful evaluation standards—e.g., inconsistent splits, unrealistic settings, or lack of competitive non-GNN baselines. Our call is not only for new datasets alone but also for restructuring how benchmarks are designed and evaluated in these domains.
>
>
> While we appreciate the reviewer’s concerns, our paper makes a necessary argument for rethinking graph benchmarks to encourage engagement beyond the graph ML community. We refine our framing to emphasize that our position is not only about dismissing current applications but improving how they are benchmarked to ensure real-world relevance.
>
> **Please consider updating your score if you are satisfied with our answer. We are happy to answer any remaining questions.**

---

> > ### Comment · Reviewer_AfXm · 2025-04-05
> >
> > Thank you for the response, I have updated the score accordingly.

---

> > > ### Author Response · Authors · 2025-04-05
> > >
> > > Thank you, please let us know if you have any remaining concerns.

---

### Official Review · Reviewer_4zxs · 2025-03-16

**Significance:** 3
**Argument Clarity:** 4
**Rating:** 4
**Confidence:** 4

**Questions:**

See “critical discussion” section.

**Discussion Potential:**

4

**Paper Summary:**

Thanks to the authors for taking the time to put together this manuscript. I enjoyed reading it and agree with much of what was written.

Summary:

This paper discusses the slow down in the graph machine learning community, attributing the malaise to failures to establish adequate benchmarks that address important problems of real significance and are clearly dominated by GNN based methods.

These problems have been discussed for quite some time in the community, so in a sense this position paper does not offer a radial new opinion. Instead, this position paper organizes several strands of this thought into one place, providing a meeting point for graph ML researchers to discuss these ongoing problems that are usually discussed verbally. I am confident that this paper will lead to valuable discussions within the community and am confident in recommending its acceptance.

**Position:**

Yes

**Position In Title:**

Yes

**Related Work:**

4

**Strengths And Weaknesses:**

> Strengths

- Basic position is sensible and reflects a widely held opinion
- The graph ML community needs to reckon with these problems more, and this paper helps stimulate that discussion.
- Despite the basic position of the paper being “obvious”, the discussion provides a good entry point to this important debate, and lays out several key elements of the situation, including highly unrealistic benchmarks and a reality check on architecture progress

> Weaknesses

Instead of “weaknesses” I would like to offer a “critical discussion” section, containing some of my personal opinions on the topic. My hope is that the authors have some reaction to these thoughts, and the rebuttal discussion can in some way contribute to the position paper.

---

Whilst I agree with your points on benchmark quality, my core criticism of this position paper is that I don’t find the proposed re-think is sufficiently radical, and worry that it will not be enough to address the communities’ problems. The basic idea of this paper still seems to be “there should be graph benchmarks, we just need to make them be good”, where “good” means they represent real and important problems.

I wonder if we need to flip this logic on its head. Instead of “good graph benchmarks”, there should just be “good benchmarks” - i.e., interesting problems that someone (not a graph person) wants to solve. These could be in the domains you suggest, like combinatorial optimization, but the key point is not to create just another graph community benchmark, but instead go and meet the existing combinatorial optimization community where they are and try to solve problems they recognize as interesting. This will force researchers to pit their graph solutions against non-graph methods. This way graph ML can be what it really is - a “methods-centric” topic - and if it can compete in this setting then we have a real win for graph ML.

As long as we continue to have within-group benchmarks that only graph people care about, we will not have fully escaped our benchmark problem. The only way to guarantee we have achieved this is to make sure we find (or develop) benchmarks to that non-graph people also care about, since this forces us to compare GNNs against non-graph approaches.

I know that in Figure 1 you say “Benchmarks should include baselines with unstructured sets”. Another way of putting my point is to say that when you say “unstructured baselines” it makes me think of MLPs or DeepSets or other very simple/in-group methods. To be truly compelling, we cannot simply have “unstructured baselines” - we must actually be competing against non-graph methods.

However this is a potentially brave new world for graph researchers, as we would have to actually go out of our comfort zone. This may be inevitable, as we come to realize that a method-centric community cannot sustain good benchmarking culture in isolation. Neither NLP nor Vision are method centric, which is why they survive the comings and goings of specific ML methods. If Graph ML is to somehow shift to be more problem-centric then the researchers themselves  must think more seriously about the domains in which they are working (as opposed to merely “collaborating” more closely with domain experts, as recommended in this paper).

To truly do that we may need to become actual domain experts, at least to a degree, in a graph-adjacent problem-centric field.  It’s an exciting time - there are so many interesting highly structured problems that members of the graph community can and are having a big impact on. Maybe the call to the graph ML field should be to find an application domain and really learn something about it, and try to be a champion of graph ML in that field, and show that graph ML methods can solve problems in that field. If many members of the graph ML community took this approach, then we can launch a breadth-first-search across problem-centric fields, hoping to find a place from graph ML.

**Support:**

4

---

> ### Author Rebuttal · Authors · 2025-03-27
>
> > Whilst I agree with your points on benchmark quality, my core criticism of this position paper is that I don’t find the proposed re-think is sufficiently radical, and worry that it will not be enough to address the communities’ problems. The basic idea of this paper still seems to be “there should be graph benchmarks, we just need to make them be good”, where “good” means they represent real and important problems.
>
> > I wonder if we need to flip this logic on its head. Instead of “good graph benchmarks”, there should just be “good benchmarks” - i.e., interesting problems that someone (not a graph person) wants to solve. These could be in the domains you suggest, like combinatorial optimization, but the key point is not to create just another graph community benchmark, but instead go and meet the existing combinatorial optimization community where they are and try to solve problems they recognize as interesting. This will force researchers to pit their graph solutions against non-graph methods. This way graph ML can be what it really is - a “methods-centric” topic - and if it can compete in this setting then we have a real win for graph ML.
>
> > As long as we continue to have within-group benchmarks that only graph people care about, we will not have fully escaped our benchmark problem. The only way to guarantee we have achieved this is to make sure we find (or develop) benchmarks to that non-graph people also care about, since this forces us to compare GNNs against non-graph approaches.
>
> > I know that in Figure 1 you say “Benchmarks should include baselines with unstructured sets”. Another way of putting my point is to say that when you say “unstructured baselines” it makes me think of MLPs or DeepSets or other very simple/in-group methods. To be truly compelling, we cannot simply have “unstructured baselines” - we must actually be competing against non-graph methods.
>
> > However this is a potentially brave new world for graph researchers, as we would have to actually go out of our comfort zone. This may be inevitable, as we come to realize that a method-centric community cannot sustain good benchmarking culture in isolation. Neither NLP nor Vision are method centric, which is why they survive the comings and goings of specific ML methods. If Graph ML is to somehow shift to be more problem-centric then the researchers themselves must think more seriously about the domains in which they are working (as opposed to merely “collaborating” more closely with domain experts, as recommended in this paper).
>
> > To truly do that we may need to become actual domain experts, at least to a degree, in a graph-adjacent problem-centric field. It’s an exciting time - there are so many interesting highly structured problems that members of the graph community can and are having a big impact on. Maybe the call to the graph ML field should be to find an application domain and really learn something about it, and try to be a champion of graph ML in that field, and show that graph ML methods can solve problems in that field. If many members of the graph ML community took this approach, then we can launch a breadth-first-search across problem-centric fields, hoping to find a place from graph ML.
>
> We thank you for your detailed, constructive review and suggestions for further improvements.
>
> That is a good point - devising benchmarks interesting to both the graph learning community and domain experts, e.g., combinatorial optimization, is challenging. However, we will account for your comments in the next revision and stress this point more explicitly.
>
> Additionally, while we agree that NLP and vision are problem-driven fields, they still rely on methodologically sound benchmarks that the community curates. We argue that graph ML should follow a similar trajectory—developing useful benchmarks beyond the graph ML community while maintaining rigorous evaluation practices.
> We acknowledge that simply including unstructured baselines is insufficient; we must actively compare against domain-relevant, non-graph approaches. We are already moving in this direction by proposing incorporating diverse baselines (including those used by domain experts) and advocating for benchmarks that reflect real-world tasks where graph ML must prove its value.
> We also agree that more profound domain expertise is valuable — if not necessary — and encourage researchers to immerse themselves in application areas. However, we view this as complementary to, rather than replacing, a principled benchmarking framework that supports meaningful cross-domain evaluation.
> In summary, our work lays the groundwork for the reviewer's transition: it provides a structure that allows graph ML to step outside its bubble, fostering rigorous comparisons and deeper problem-centric engagement.

---

### Decision · Program_Chairs · 2025-04-30

**Decision:**

Accept (poster)

**Comment:**

All reviewers agree that this position paper makes important contributions to the advancement of graph machine learning. It provides sufficient empirical results to support its claims and proposes reasonable solutions to address existing limitations in the field. A few minor concerns were also raised, but these have been addressed during the rebuttal phase. It is encouraged to incorporate these clarifications into the final version to further strengthen the submission. The alternative views section needs to be expanded to clearly state the alternatives and respond to them.